# The Association Between Identity Functioning and Personality Pathology in Female Patients with Eating Disorders

**DOI:** 10.3390/nu17142329

**Published:** 2025-07-16

**Authors:** Laurence Claes, Annabel Bogaerts, Tim Bastiaens, Glenn Kiekens, Eva Dierckx, Katrien Schoevaerts, Koen Luyckx

**Affiliations:** 1Faculty of Psychology and Educational Sciences, KU Leuven, 3000 Leuven, Belgiumkoen.luyckx@kuleuven.be (K.L.); 2Faculty of Medicine and Health Sciences, University Antwerp, 2000 Antwerp, Belgium; 3Department of Psychology, Clinical Developmental Psychology Program, University of Amsterdam, 1001 Amsterdam, The Netherlands; a.bogaerts@uva.nl; 4Campus Kortenberg, University Psychiatric Centre, KU Leuven, 3070 Kortenberg, Belgium; tim.bastiaens@upckuleuven.be; 5Department of Neurosciences, Center for Contextual Psychiatry, KU Leuven, 3000 Leuven, Belgium; 6Department of Medical and Clinical Psychology, Tilburg University, 5037 Tilburg, The Netherlands; 7Psychiatric Hospital Alexianen Zorggroep Tienen, 3300 Tienen, Belgiumkatrien.schoevaerts@azt.broedersvanliefde.be (K.S.); 8Department of Clinical Psychology, Vrije Universiteit Brussel, 1050 Brussels, Belgium; 9Educational Unit for Professional Training and Service in the Behavioural Sciences (UNIBS), University of the Free State, Bloemfontein 9301, South Africa

**Keywords:** identity, personality disorders, eating disorders

## Abstract

Aims. In the present study, we investigated the associations between the three identity dimensions of Kaufman (Consolidated Identity, Disturbed Identity, Lack of Identity) and symptoms of personality disorders (PDs) in 176 female inpatients with an eating disorder (ED). We examined five aspects: the prevalence of categorical PD diagnoses in patients with EDs; the relationship between dimensional PD scores and identity dimensions as well as their relationships with age and ED subtype; and the unique variance in dimensional PD scores explained by identity dimensions, while controlling for age and ED subtype. Methods. To assess identity functioning, we made use of the Self-Concept and Identity Measure, and to assess PDs, we used the categorical and dimensional scores of the Assessment of DSM-IV Personality Disorders. Results. The findings showed that the avoidant, obsessive–compulsive, and borderline categorical PDs were the most frequently reported PDs. Age was negatively related to all Cluster B PDs and Disturbed Identity, and binge-eating/purging ED patients reported significantly more Cluster B PD features compared to restrictive ED patients. ED subtype and identity dimensions were unrelated. Correlational analysis showed that all dimensional PD scores were positively related to Disturbed Identity and Lack of Identity and negatively related to Consolidated Identity. The results of the hierarchical regression analyses showed that Cluster A PDs were significantly predicted by Lack of Identity, controlled for age and ED subtype. Additionally, Cluster B PDs were significantly predicted by Disturbed Identity. Finally, two of the three cluster C PDs were predicted by Lack of Identity (avoidant and obsessive–compulsive PD), whereas the dependent PD was explained by Disturbed Identity. Conclusions. The co-occurrence of identity issues in both PDs and EDs underscores the role of identity as a transdiagnostic feature. Accordingly, using identity-based interventions in treatment may have broad therapeutic benefits across these disorders.

## 1. Introduction

Understanding the associations between identity functioning and personality disorders (PDs) in eating disorders (EDs) offers critical insights into the co-occurrence of these complex conditions. Identity disturbance not only underpins many of the core features of EDs but also serves as a bridge linking EDs to co-occurring personality pathology. In the present study, we will investigate the prevalence of categorical PDs in patients with EDs, examine the associations between identity functioning and dimensional PD scores as well as their relationships with age and ED subtype, and research the unique variance that identity dimensions can explain in dimensional PD scores (while controlling for age and ED subtype).

### 1.1. Identity

In Erikson’s [1] influential lifespan theory, identity formation is conceptualized as a dynamic tension between identity synthesis and identity confusion. Successful development requires individuals to achieve a balance that favors synthesis over confusion. Identity synthesis reflects the extent to which different aspects of the self are integrated, contributing to a subjective sense of continuity and coherence across time and situations. In contrast, identity confusion is characterized by difficulties in making and sustaining long-term commitments, as well as a lack of clear purpose and direction in life [1,2,3].

To advance the study of identity in both community and clinical populations, Kaufman et al. [4] proposed a multidimensional model of identity, grounded in both developmental and clinical theoretical frameworks [1,5]. This model delineates three core dimensions of identity functioning. (1) Consolidated Identity reflects a stable and coherent sense of self, characterized by a clear understanding of who one is, commitment to personal values and beliefs, consistency across time and situations, and a sense of positive self-worth [4]. (2) Disturbed Identity captures disruptions in this coherence, including internal inconsistencies in values, beliefs, and opinions, as well as an overreliance on external sources—such as others’ expectations or approval—to define the self [4]. (3) Lack of Identity, a dimension that has received comparatively less empirical attention, refers to a profound sense of inner emptiness and disconnection of the self. Individuals scoring high on this dimension often report feeling lost, broken, or devoid of a coherent self-concept, with little or no sense of who they are [4]. This tripartite framework offers a nuanced understanding of identity functioning and its impairments, with relevance for both normative development and psychopathology. To assess the three identity dimensions, Kaufman et al. [4] developed the Self-Concept and Identity Measure (SCIM), a reliable and valid questionnaire that has demonstrated convergent validity with measures of borderline PD, emotion dysregulation, and various other indicators of psychopathology [4,6].

### 1.2. Identity and Eating Disorders

Bruch [7,8] proposed that EDs, particularly anorexia nervosa, arise from fundamental disturbances in identity development and autonomy, rather than purely sociocultural ideals. Contemporary perspectives support this view, highlighting maladaptive identity functioning as central across restrictive and binge/purge EDs [3,9]. In restrictive EDs, individuals often exhibit a poorly differentiated self, linking identity to control over food and body [10,11,12]. The pursuit of thinness serves as a compensatory strategy for fragmented identity and perceived ineffectiveness [7,8,10,11]. In binge/purge EDs, individuals experience chronic emptiness, using bingeing to fill emotional voids and purging to regain control, reflecting an unstable self-concept and impaired emotion regulation [3,13,14,15].

Numerous empirical studies have shown that individuals with eating disorders (EDs) exhibit greater identity confusion, and that maladaptive identity functioning is positively correlated with ED symptom severity [14,16,17,18,19]. Verschueren et al. [20] found that ED patients scored lower on adaptive and higher on maladaptive identity processes compared to controls, identifying an “identity disorder” status exclusively among ED patients. Although many studies have examined the link between identity and EDs, the directionality of this relationship has been less explored. In a two-year longitudinal study, Verschueren et al. [21] demonstrated that identity confusion increased vulnerability to body dissatisfaction and bulimia symptoms, while identity synthesis was protective. Moreover, bulimia symptoms predicted increased identity confusion and decreased identity synthesis, highlighting a bidirectional relationship. Recent findings further show that reductions in identity confusion and increases in identity synthesis during treatment are associated with decreases in drive for thinness and body dissatisfaction [22]. These results suggest that recovery involves relinquishing an ED-centered identity and developing a more integrated self [23,24,25]. Finally, Rohde et al. [18] found that patients with bulimia nervosa exhibited greater identity impairment than those with restrictive-type anorexia nervosa.

### 1.3. Identity and Personality Disorders in Patients with Eating Disorders

Given the high prevalence of identity disturbances among individuals with eating disorders (EDs) and their established association with personality disorders (PDs) [26], it is unsurprising that many ED patients also exhibit PD features. Most studies examining PDs in ED populations have employed the categorical approach outlined in DSM-5 Section II [27], which defines ten PDs grouped into three clusters: Cluster A (odd/eccentric: paranoid, schizoid, and schizotypal PDs), Cluster B (dramatic/emotional: narcissistic, antisocial, borderline, and histrionic PDs), and Cluster C (anxious/fearful: avoidant, dependent, and obsessive–compulsive PDs). Approximately 50–70% of individuals with an ED meet criteria for at least one PD [28,29,30], with Cluster C PDs being most prevalent, followed by Cluster B and Cluster A PDs [31].

By subtype, individuals with ED binge/purge type (ED-BP) show higher rates of Cluster B PDs, especially the borderline PD, compared to restrictive-type ED (ED-R) patients [30,32]. No consistent differences have been observed for Cluster C PDs between subtypes [28,29,33]. Additionally, PD prevalence tends to decrease with age in ED populations [31].

While identity disturbance is a diagnostic criterion for the borderline PD, the Alternative Model for Personality Disorders (AMPD) in DSM-5 Section III [27] considers identity impairment a core feature of all PDs. The AMPD conceptualizes PDs dimensionally, defined by impairments in self-functioning (identity, self-direction) and interpersonal functioning (empathy, intimacy) (Criterion A), along with maladaptive personality traits (Criterion B).

To date, only two studies have examined identity impairments—as conceptualized by Kaufman [4]—and their relationship with dimensional scores of PDs in clinical populations. First, Bogaerts et al. [34] investigated this association in a sample of 153 psychiatric inpatients, using the Self-Concept and Identity Measure (SCIM) [4]. Their findings indicated that Lack of Identity was not significantly associated with age, Disturbed Identity was negatively correlated with age, and Consolidated Identity was positively associated with age. Correlation analyses revealed that Consolidated Identity was negatively associated with all PDs, except the narcissistic and obsessive–compulsive PDs. Disturbed Identity was positively associated with all PDs except the schizoid PD, while Lack of Identity was positively related to all PDs except the narcissistic PD. Regression analyses further demonstrated that Consolidated Identity did not significantly predict variance in any PDs other than the narcissistic PD. In contrast, Disturbed Identity was a significant positive predictor of all PDs except the schizoid and avoidant PDs. Lastly, Lack of Identity significantly predicted variance in all Cluster A PDs, as well as the borderline, avoidant, and dependent PDs.

Second, Tressova et al. [35] examined the bivariate correlations between the SCIM subscales and personality pathology as defined in Section III of the DSM-5, in a sample of 92 male forensic patients and 139 healthy controls. Disturbed Identity and Lack of Identity were positively associated with all six assessed personality disorders—antisocial, avoidant, borderline, narcissistic, obsessive–compulsive, and schizotypal PD. In contrast, Consolidated Identity was not significantly associated with any of the PDs.

### 1.4. The Present Study

The present study aimed to examine the prevalence of categorical PDs in patients with EDs and was the first to investigate the associations between identity functioning, as conceptualized by Kaufman et al. [4], and dimensional PD symptom scores in individuals with EDs. Specifically, this study pursued five main objectives. First, we assessed the prevalence of categorical PD diagnoses among individuals with EDs. Drawing on prior research [29], we hypothesized the highest prevalence rates for obsessive–compulsive, avoidant, borderline, dependent, and paranoid PDs. Furthermore, we anticipated that Cluster C PDs would be the most frequent, followed by Cluster B, with Cluster A PDs being the least prevalent [31].

Second, we investigated whether age and ED subtype—restrictive (ED-R) versus binge/purge (ED-BP)—were associated with dimensional symptom scores of PDs. Prior research in ED populations has demonstrated significantly negative associations between age and PD symptoms, with younger individuals typically exhibiting more pronounced PD traits [31]. Within Cluster B, we expected ED-BP patients to exhibit higher levels of borderline PD symptoms compared to those with ED-R. In contrast, we anticipated either no significant differences within Cluster C or a slightly higher prevalence of obsessive–compulsive PD among ED-R patients [30,32].

Third, we explored associations between age, ED subtype, and Kaufman’s et al. [4] three dimensions of identity functioning: Lack of Identity, Disturbed Identity, and Consolidated Identity. Based on previous findings [34], we hypothesized that age would be positively associated with Consolidated Identity and negatively associated with Disturbed Identity, while no significant association was expected with Lack of Identity. Guided by the findings of Rohde et al. [18], we further anticipated that patients engaging in binge-eating/purging behaviors would demonstrate higher levels of identity-related deficits, compared to those with restrictive behaviors.

Fourth, we examined associations between the three identity dimensions and the dimensional symptom scores of PDs, by controlling for age and ED subtype. Consistent with the existing literature [34,35], we expected that both Disturbed Identity and Lack of Identity would show positive associations with most PDs, except the narcissistic PD [34]. In contrast, we hypothesized that Consolidated Identity would be negatively associated with all PDs (except the narcissistic and obsessive–compulsive PD) [34] or would be unrelated to all PDs [35].

Fifth, we investigated the unique contribution of each identity dimension to the variance in dimensional symptoms scores of PDs while controlling for age and ED subtype. Specifically, we expected that Cluster A PDs would be primarily predicted by Lack of Identity, whereas Cluster B PDs would be significantly associated with Disturbed Identity [34]. Given that the borderline PD is characterized by Disturbed Identity and chronic feelings of emptiness (as described in DSM-5, Section II), we also expected that Lack of Identity, related to feelings of emptiness, would further predict variance in the borderline PD [34]. Moreover, considering the DSM-5 criteria for narcissistic PD—marked by grandiosity and an inflated sense of self—we hypothesized that Consolidated Identity would predict additional variance in the narcissistic PD, and potentially in the antisocial PD as well [34]. Finally, we hypothesized that the dependent and obsessive–compulsive PDs (Cluster C) would be uniquely predicted by Disturbed Identity, while the avoidant PD would be uniquely predicted by Lack of Identity. This expectation was based on findings by previous research [34,36] and the well-documented high comorbidity between the avoidant and Cluster A PDs [37].

## 2. Materials and Methods

### 2.1. Participants and Procedure

Data were collected from the clinical records of female inpatients treated at a specialized unit for EDs in Flanders (Belgium). The current sample included 176 female inpatients (age ≥ 18 years), of whom 46 (26.1%) were diagnosed as AN-R, 16 as AN-BP (9.1%), 23 as BN (13.1%), 8 (4.5%) as BED, and 83 (47.2%) as EDNOS by means of the EDI-3 Symptom Checklist (EDI-3 SC) [38]. The mean age of the patients was 25.23 (*SD* = 6.91; range: 18–48 years). Regarding the highest level of education, 5 (2.8%) patients had completed primary education, 50 (28.4%) secondary education, and 121 (68.8%) higher education. Given that personality functioning is often related to ED subtype, we divided our ED sample into a restrictive (ED-R; N = 96, 54.5%) and a binge-eating/purging (ED-BP; N = 80, 45.5%) subgroup. The mean BMI of both groups was significantly different [*F*(1, 173) = 24.58, *p* < 0.001, partial ŋ2 = 0.13], with restrictive patients (*M*_BMI_ = 16.31, *SD*_BMI_ = 2.53) reporting a significantly lower BMI than patients with ED-BP (*M*_BMI_ = 19.17, *SD*_BMI_ = 4.91). The mean age difference between both groups was borderline-significant [*F*(1, 174) = 3.94, *p* = 0.049, partial ŋ2 = 0.02], with restrictive patients being slightly younger (*M*_Age_ = 24.29, *SD*_Age_ = 6.78) than patients with ED-BP (*M*_Age_ = 26.35, *SD*_Age_ = 6.94). Importantly, since the sample consisted exclusively of female patients with EDs, no conclusions can be drawn regarding male patients.

Patients gave written informed consent and permission to the pseudonymized use of their data for research purposes by signing an informed consent form at admission and before filling out the questionnaires on the PC. This procedure was approved by the medical–ethical committee of the clinic and the Social and Societal Ethics Committee of KU Leuven. Patients were not compensated for their participation in this study.

### 2.2. Instruments

All ED patients completed the Self-Concept and Identity Measure (SCIM [4]; Dutch version: [34,39]) and the Assessment of DSM-IV Personality Disorders (APD-IV [40,41]) to assess identity functioning and PD symptomatology.

The Self-Concept and Identity Measure (SCIM [4]; Dutch translation: [34,39]) comprises 27 items, each rated on a 7-point Likert scale ranging from 1 (“completely disagree”) to 7 (“completely agree”). The SCIM yields three subscales: Consolidated Identity (10 items, e.g., “I know who I am”), Disturbed Identity (11 items, e.g., “I change a lot depending on the situation”), and Lack of Identity (6 items, e.g., “I feel empty inside, like a person without a soul”). Subscale scores are calculated as the mean score of the corresponding item scores. The Dutch version of the SCIM has been validated in both the general population and clinical samples [34,39]. In the clinical sample, confirmatory factor analysis supported the three-factor model of the SCIM, with a good model fit achieved after the exclusion of items 3, 11, and 14 from the Consolidated Identity Scale and item 23 from the Disturbed Identity scale [34]. In the present study, the internal consistency (Cronbach’s alpha) for the revised subscales was as follows: α = 0.68 for Consolidated Identity (7 items), α = 0.81 for Disturbed Identity (10 items), and α = 0.86 for Lack of Identity (6 items). According to George and Mallery [42], alpha values above 0.70, 0.80, and 0.90 indicate acceptable, good, and excellent reliability, respectively. The Disturbed Identity and Lack of Identity Scales demonstrated good reliability coefficients, while the Consolidated Identity Scale exhibited marginally acceptable reliability. These results align with previous studies employing the Dutch version of the SCIM [34,39]. Excluding the item “I am basically the same person that I’ve always been” from the Consolidated Identity Scale would have increased the alpha coefficient to α = 0.72, indicating acceptable reliability. However, to ensure comparability with prior findings [34,39], this item was retained.

The Assessment of DSM-IV Personality Disorders (ADP-IV [40,41]) was used to assess personality disorder symptomatology. The ADP-IV is a Dutch self-report questionnaire consisting of 94 items, covering the diagnostic criteria for the 12 personality disorders (PDs) defined in the DSM-IV [43]. The structure of the ADP-IV allows for both dimensional trait scoring and categorical PD diagnosis. Each item (e.g., a borderline PD item: “I absolutely cannot bear the idea that someone would leave or abandon me; therefore, I will do anything to prevent this”) is rated on a 7-point Likert scale, ranging from 1 (“totally disagree”) to 7 (“totally agree”), yielding a trait score. If a trait score is 5 (“rather agree”) or higher, respondents are further asked to indicate the degree of distress the feature causes themselves or others, using a 3-point scale (1 = “not at all”; 3 = “most certainly”). Mean dimensional trait scores are calculated by summing the trait scores and dividing by the number of items corresponding to each specific PD. Categorical diagnoses are derived according to the DSM-IV criteria, using combined trait and distress scores within validated scoring algorithms (e.g., a criterion is considered present if Trait > 4 and Distress > 1) [40,41]. In the present study, both categorical diagnoses and dimensional trait scores were used. Internal consistency (Cronbach’s alpha) for the ADP-IV dimensional scales ranged from 0.68 (marginally acceptable reliability; the antisocial PD) to 0.81 (good reliability; the schizotypal PD). Excluding the item “I experience little or no feelings of guilt or remorse when I have done something wrong” from the Antisocial PD Scale would have increased the alpha coefficient to α = 0.71, indicating acceptable reliability. Nevertheless, this item was preserved to allow for direct comparison with earlier studies [34,39].

### 2.3. Analyses

All analyses were performed by means of the program SPSS, version 29. First, to determine the prevalence of the categorical PDs in our ED sample, we calculated the number (percentages) of the patients who met the diagnostic criteria of the 10 categorical PDs. Second, we calculated the descriptive statistics of the dimensional symptom scores of PDs. To investigate the association between the dimensional symptom scores of PDs and age, we calculated the Pearson correlation coefficients. To investigate whether the dimensional symptoms scores of PDs differed between ED subtype (ED-R vs. ED-BP), we performed a MANOVA with the dimensional symptom scores of PDs as dependent variables and ED subtype as independent variable. Third, to investigate the descriptive statistics of the three SCIM identity scales, we calculated the means and standard deviations of the three scales. Associations between the SCIM scales and age were calculated by means of the Pearson correlation coefficients. To investigate differences in the three SCIM scales between ED subtype, we performed a MANOVA with the SCIM scales as dependent variables and ED subtype as independent variable. Fourth, to determine the associations between the three SCIM identity scales and the dimensional symptom scores of PDs (controlled for age and ED subtype depending on the outcomes of the prior analyses), we calculated partial correlations. Hemphill [44] suggests that correlation coefficients below 0.20, between 0.20 and 0.30, and above 0.30 indicate small, medium, and large effects, respectively. Finally, to investigate which SCIM scales predicted the most variance in each of the 10 dimensional symptom scores of PDs, while controlling for age and ED subtype, we performed 10 hierarchical regression analyses with each of the dimensional symptom scores of PDs as dependent variables, and in step 1 (age) and step 2 (ED subtype) the control variables, and in step 3 (the three SCIM scales) the predictors.

## 3. Results

### 3.1. Prevalence of Categorical PDs in Patients with EDs

The prevalence rates of the categorical PDs in our ED sample are displayed in Table 1. The obsessive–compulsive PD (66.5%), the avoidant PD (58.8%), and the borderline PD (46%) were the most frequently reported PDs, followed by the dependent PD (23.3%) and the Cluster A PDs.

### 3.2. Associations Between Dimensional Symptom Scores of PDs and Age and ED Subtype

The means (standard deviations) of the dimensional symptom scores of PDs are shown in Table 1. Age showed a significantly negative association with the dimensional scores of all Cluster B PDs and the dimensional score of the dependent PD, with a medium effect size. Regarding ED subtype, ED-BP patients reported significantly higher dimensional scores for the antisocial and the histrionic PD (Cluster B) compared to restrictive patients.

### 3.3. Associations Between Dimensional SCIM Identity Scale Scores and Age and ED Subtype

The mean scores on the three SCIM identity scale scores are displayed in Table 1, and the highest mean score is indicated for Lack of Identity (*M* = 4.52; 4 = neither agree nor disagree, 5 = somewhat agree), followed by Consolidated Identity (*M* = 4.19; 4 = neither agree nor disagree) and Disturbed Identity (*M* = 3.21; 3 = somewhat disagree). Age showed a significant negative association with Disturbed Identity (medium effect size), while no associations were found with Lack of Identity and Consolidated Identity. Finally, no significant differences emerged between the two ED subtypes (ED-R vs. ED-BP) on the three SCIM scales.

### 3.4. Correlations Between SCIM Identity Scale Scores and Dimensional Symptom Scores of PDs Controlled for Age and ED Subtype

The partial correlations between the three SCIM identity scales and the dimensional PD scales (controlled for age and ED subtype (ED-R vs. ED-BP); Table 2) showed that all PDs were significantly positively related to Disturbed Identity and Lack of Identity (except the narcissistic and histrionic PDs), and significantly negatively related to Consolidated Identity (except the narcissistic, histrionic, and obsessive–compulsive PDs). The strongest positive correlations (large effect sizes) were observed between Disturbed Identity and the Cluster A and Cluster B PDs, as well as between Lack of Identity and the Cluster A PDs, avoidant PD, and borderline PD. Finally, the strongest negative correlations were found between Consolidated Identity and the Cluster A PDs and avoidant PD.

### 3.5. Prediction of Dimensional Symptom Scores of PDs by Means of SCIM Identity Scale Scores Controlled for Age and ED Subtype

The results of the hierarchical regression analyses with the dimensional PD scales as dependent variables are displayed in Table 3. Overall, the findings indicate that adding the three identity scales explained a significant amount of additional variance in all dimensional PD scores, above and beyond the effects of age and ED subtype. The largest increases in explained variance were observed for the Cluster A paranoid PD (ΔR^2^ = 0.263), schizotypal PD (ΔR^2^ = 0.226), and histrionic PD (ΔR^2^ = 0.245). When examining the total variance explained across all PDs by age, ED subtype, and identity dimensions, the proportion of explained variance ranged from R^2^ = 0.100 (obsessive–compulsive PD) to R^2^ = 0.360 (histrionic PD). These results suggest that there remains substantial room for improving the prediction of dimensional PDs beyond age, ED subtype, and identity dimensions.

Concerning Cluster A PDs, Lack of Identity positively predicted the dimensional scores of all three Cluster A PDs (paranoid, schizoid, and schizotypal), while controlling for age, ED subtype, and the other SCIM scales. Regarding Cluster B PDs, age was significantly negatively related to the dimensional scores of all Cluster B PDs (step 1), indicating that Cluster B PDs are more frequently reported by younger female patients with an ED. After controlling for age, ED subtype also significantly predicted the dimensional scores of three out of four Cluster B PDs (excluding the narcissistic PD), suggesting that Cluster B PDs are more prevalent in patients with binge-eating/purging-type EDs compared to restrictive-type EDs. Furthermore, Disturbed Identity positively predicted unique variance in all dimensional Cluster B PDs, after controlling for age and ED subtype. Additionally, the dimensional borderline PD was also positively predicted by Lack of Identity, and the narcissistic PD by Consolidated Identity. Finally, for Cluster C PDs, the dependent PD was significantly positively predicted by Disturbed Identity, whereas the avoidant and obsessive–compulsive PDs were significantly predicted by Lack of Identity, in line with the findings for Cluster A PDs.

## 4. Discussion

Given the high comorbidity between eating disorders (EDs) and personality disorders (PDs) [28,29,30], it may be crucial to consider identity dysfunction as a potential transdiagnostic mechanism underlying this co-occurrence. This perspective aligns with contemporary dimensional models of psychopathology, such as the Research Domain Criteria (RDoC) framework proposed by the National Institute of Mental Health [45]. This model emphasizes the importance of shared underlying psychological processes—such as impairments in self and identity functioning—that surpass traditional diagnostic boundaries. Identity dysfunction, characterized by difficulties in establishing a coherent and stable sense of self, has been increasingly recognized as a core feature in both EDs and PDs [4,34]. As such, conceptualizing identity dysfunction as a transdiagnostic vulnerability factor may offer a more integrative understanding of the high comorbidity between these disorders and inform more targeted interventions.

### 4.1. Prevalence of Categorical PDs in Patients with EDs

In line with prior research [29], we found that the obsessive–compulsive, avoidant, and borderline categorical PDs were the most prevalent categorical PDs among individuals with EDs. Overall, Cluster C PDs were most prevalent, followed by Cluster B and Cluster A PDs, respectively [31]. This pattern suggests that anxiety-related traits (Cluster C) and emotional dysregulation (Cluster B) are particularly common among ED patients [28,29].

### 4.2. Associations Between Dimensional Symptom Scores of PDs and Age and ED Subtype

In line with previous research, we observed significant negative associations between age and dimensional scores of all Cluster B PDs, as well as the dependent PD [31,37,46]. The decline in dependent traits with age may reflect increased autonomy and social maturity over time [47]. Similarly, reduction in Cluster B traits is consistent with findings indicating that impulsivity and affective instability typically decrease with age [48]. Furthermore, patients with binge-eating/purging behaviors showed higher Cluster B PD scores compared to those with restrictive eating behaviors. This supports evidence that impulsivity—a core feature of Cluster B PDs—is more pronounced in ED subtypes involving bingeing and purging, emphasizing the link between behavioral dysregulation and identity disturbances in these groups [28,49].

### 4.3. Associations Between Dimensional SCIM Identity Scale Scores and Age and ED Subtype

As hypothesized and in line with prior findings [34], Disturbed Identity was negatively associated with age, indicating some normative improvement in identity integration over time. However, Lack of Identity was unrelated to age, and contrary to our expectations, Consolidated Identity was not significantly related to age. While identity consolidation typically increases in community samples [20], this trend may be disrupted in clinical ED samples, where chronic psychopathology can hinder the development of a coherent and synthesized identity. Unlike Rohde et al. [18], we did not observe significant differences in SCIM dimensions between ED subtypes (ED-R vs. ED-BP). This divergence may be due to differences in subgroup definitions: our ED-BP group included both BN and AN-BP patients, possibly diluting subtype-specific effects.

### 4.4. Correlations Between SCIM Identity Scale Scores and Dimensional Symptom Scores of PDs Controlled for Age and ED Subtype

In line with earlier research, Disturbed Identity and Lack of Identity were significantly positively associated with nearly all dimensional scores of PDs, except the narcissistic and histrionic PDs, which were not significantly correlated with Lack of Identity. Conversely, Consolidated Identity was negatively associated with most dimensional scores of PDs, supporting its role as a protective factor [4,34]. These findings support the importance of identity dysfunction as a core diagnostic criterion of most PDs, as recognized in the Alternative Model for Personality Disorders of the DSM-5 [50,51].

### 4.5. Prediction of Dimensional Symptom Scores of PDs by Means of SCIM Identity Scale Scores Controlled for Age and ED Subtype

Finally, when predicting PD symptoms controlled for age and ED subtype, Lack of Identity significantly predicted all Cluster A PDs (paranoid, schizoid, and schizotypal) and two Cluster C PDs (avoidant and obsessive–compulsive). These PDs are characterized by impaired identity integration. In Cluster A PDs, identity is often fragmented or unintegrated, resulting in bizarre thoughts, social withdrawal, and suspiciousness. As individuals with Cluster A PDs often present with a psychotic personality structure [5,52], their reality testing and sense of identity may be compromised, leading to blurred self-other boundaries and a fragmented self [34]. In Cluster C PDs, patients tend to be overly contingent, either on others’ approval (avoidant PD) or rigid performance standards (obsessive–compulsive PD).

All dimensional Cluster B PDs scores were predicted by Disturbed Identity, typically expressed through a fluctuating self-image, externalized self-worth, and fragmented internal cohesion [53,54]. Notably, the borderline PD was predicted by both Disturbed Identity and Lack of Identity. This aligns with the literature highlighting chronic feelings of emptiness and dissociative states marked by feelings of nothingness, numbness, and disconnection from self and others as central to borderline pathology [34,53,55]. The narcissistic PD also involves both disturbed and partially integrated identity structures. In grandiose narcissism, individuals often maintain a highly idealized and inflated self-image; however, this self-concept is fragile, defensive, and dependent on external validation, achievement, and dominance, making it vulnerable to destabilization by criticism or failure [34,56].

Finally, the dependent PD was also predicted by Disturbed Identity alone, suggesting a failure in the individuation processes, resulting in an identity excessively reliant on close relationships and vulnerable to interpersonal disruptions [47].

When considering age, ED subtype, and identity dimensions, the proportion of explained variance in PD scores ranged from R^2^ = 0.100 (obsessive–compulsive PD) to R^2^ = 0.360 (histrionic PD). This indicates that substantial variance remains unexplained, highlighting the need to include additional factors. According to the Alternative Model for Personality Disorders (AMPD) in DSM-5 Section III [50,51], personality disorders arise from impairments in self and interpersonal functioning (Criterion A) and specific maladaptive personality traits (Criterion B). While our study focused on self-functioning through identity dimensions, incorporating measures of interpersonal dysfunction and personality traits could further improve the prediction and understanding of PD symptoms in ED patients.

### 4.6. Clinical Implications

The co-occurrence of identity disturbance in both PDs and EDs underscores its role as a transdiagnostic feature. Given its central role, identity disturbance can be considered a valuable treatment target in both PDs and EDs. Schema therapy (ST), for example, addresses maladaptive schemas underlying identity disturbance, helping patients develop a more integrated and stable sense of self; it has shown efficacy for both PDs and chronic EDs [57,58]. Mentalization-based treatment (MBT), originally developed for BPD, focuses on improving reflective functioning and self-understanding, and has been adapted successfully for EDs to enhance self-structure and emotional regulation [59]. Dialectical behavior therapy (DBT) targets identity disturbance indirectly through skills modules focused on emotion regulation, mindfulness, and interpersonal effectiveness, and has demonstrated effectiveness in patients with comorbid EDs and BPD [60,61]. By explicitly addressing identity disturbance, these approaches provide promising avenues for targeting the shared vulnerabilities underlying both PDs and EDs.

### 4.7. Limitations and Suggestions for Future Research

The present cross-sectional study examining the association between identity dimensions and dimensional scores of PDs in female patients with an ED via self-report measures presents several limitations that warrant consideration. These limitations pertain to the sample and the methodological constraints inherent in self-report designs and the cross-sectional nature of the study, which collectively impact the interpretability and generalizability of the findings. The exclusive focus on a female inpatient sample limits the generalizability of the findings to male patients and individuals receiving outpatient treatment. Additionally, EDNOS represents a heterogeneous category, and collapsing it into restrictive versus binge/purge groups may mask important clinical differences. Furthermore, we did not have access to patients’ medical files and therefore lacked information on detailed sociodemographic variables as well as physical and psychological comorbidities, which should be addressed in future studies. Finally, as most participants were White and from Western backgrounds (i.e., Flemish), the findings may not be generalizable to individuals from other ethnic, cultural, or regional groups. The reliance on self-report measures introduces susceptibility to social desirability bias and may be particularly problematic in individuals with EDs and comorbid personality disorders, who may have impaired introspective capacity [62]. The use of self-report for all study variables also raises concerns about shared method variance, potentially inflating observed associations [63]. Additionally, the reliability of the Consolidated Identity Scale and the Antisocial PD Scale was marginally adequate; however, these scales were retained in their original form to allow for direct comparison with earlier studies [34,39]. Moreover, the cross-sectional design precludes causal inference and limits insight into the temporal dynamics between identity issues and personality pathology. Future research should address these limitations by incorporating clinician-administered assessments and informant reports to reduce bias and enhance construct validity. Longitudinal designs are also recommended to elucidate the directionality of relationships. Finally, future research could also make use of assessment instruments specifically designed to measure the PD criteria outlined in the Alternative Model for Personality Disorders [64]. Such methodological improvements would strengthen the validity and clinical utility of findings in this population.

## 5. Conclusions

In sum, the co-occurrence of identity issues in both PDs and EDs underscores its role as a transdiagnostic feature. Addressing identity issues in treatment may therefore have broad therapeutic benefits across these disorders.

## Figures and Tables

**Table 1 nutrients-17-02329-t001:** Numbers (percentages) of categorical PDs, means (standard deviations) of the dimensional PD and identity scores, correlations with age, and differences in means (standard deviations) in dimensional PD and identity scores between restrictive and binge-eating/purging patients with an ED.

	Categorical PDs (*N* = 176)		Dimensional PDs (*N* = 176)							
						Restrictive (*N* = 96)		Binge/Purge (*N* = 80)		
ADP-IV	*N*	(*%*)	*M*	(*SD*)	*r*(*age*)	*M*	(*SD*)	*M*	(*SD*)	*F*
Paranoid	33	(18.8)	3.31	(1.12)	−0.05	3.30	(1.18)	3.33	(1.05)	0.045
Schizoid	31	(17.6)	3.26	(1.09)	0.03	3.24	(1.04)	3.30	(1.16)	0.138
Schizotypal	29	(16.5)	3.09	(1.08)	−0.14	3.12	(1.09)	3.05	(1.08)	0.168
Antisocial	0	(0.0)	1.75	(0.67)	−0.16 *	1.60	(0.61)	1.93	(0.69)	10.749 ***
Borderline	81	(46.0)	3.86	(1.01)	−0.23 **	3.76	(0.96)	3.99	(1.07)	2.303
Histrionic	7	(4.0)	2.95	(0.88)	−0.27 **	2.82	(0.83)	3.11	(0.91)	5.077 *
Narcissistic	2	(1.1)	2.21	(0.81)	−0.21 **	2.22	(0.81)	2.20	(0.81)	0.031
Avoidant	103	(58.8)	4.50	(1.13)	−0.06	4.50	(1.14)	4.50	(1.13)	0.001
Dependent	41	(23.3)	3.75	(0.96)	−0.24 **	3.79	(0.95)	3.70	(0.96)	0.371
Obsessive–Compulsive	117	(66.5)	4.52	(0.94)	−0.07	4.64	(0.90)	4.37	(0.97)	3.713
SCIM			*M*	(*SD*)	*r*(*age*)	*M*	(*SD*)	*M*	(*SD*)	*F*
Disturbed identity			3.21	(0.98)	−0.26 ***	3.12	(0.96)	3.31	(1.00)	1.59
Lack of Identity			4.52	(1.29)	−0.06	4.57	(1.31)	4.46	(1.26)	0.33
Consolidated identity			4.19	(0.88)	0.08	4.21	(0.86)	4.17	(0.82)	0.07

* *p* < 0.05; ** *p* < 0.01; *** *p* < 0.001; ADP-IV = Assessment of DSM-IV Personality Disorders; SCIM: Self-Concept and Identity Measure; ED = eating disorder.

**Table 2 nutrients-17-02329-t002:** Partial correlations between APD-IV and SCIM dimensional scores controlled for age and ED subtype (ED-R vs. ED-BP).

	SCIM Identity Disturbance	SCIM Lack of Identity	SCIM Consolidated Identity
ADP-IV	Partial *r*	Partial *r*	Partial *r*
Paranoid	0.35 ***	0.50 ***	−0.35 ***
Schizoid	0.16 *	0.39 ***	−0.23 **
Schizotypal	0.33 ***	0.46 ***	−0.31 ***
Antisocial	0.29 ***	0.18 *	−0.16 *
Borderline	0.33 ***	0.44 ***	−0.29 ***
Histrionic	0.52 ***	0.15	−0.15
Narcissistic	0.31 ***	0.10	0.03
Avoidant	0.22 **	0.40 ***	−0.35 ***
Dependent	0.37 ***	0.22 **	−0.26 ***
Obsessive–Compulsive	0.20 **	0.27 ***	−0.15
SCIM			
Disturbed Identity	-	0.46 ***	−0.40 ***
Lack of Identity		-	−0.62 ***
Consolidated Identity			-

* *p* < 0.05; ** *p* < 0.01; *** *p* < 0.001; ADP-IV = Assessment of DSM-IV Personality Disorders; SCIM: Self-Concept and Identity Measure; ED = eating disorder; ED-R = ED-restrictive type; ED-BP = ED-binge/purge type.

**Table 3 nutrients-17-02329-t003:** Hierarchical regression analyses with dimensional PD scores as dependent variables (ADP-IV), age and ED subtype as control variables, and identity dimensions (SCIM) as predictors.

	ADP-IV									
	PAR	SZ	ST	AS	BPD	HIS	NAR	AV	DEP	OC
	β	β	β	β	β	β	β	β	β	β
Age	−0.05	0.03	−0.14	−0.16 *	−0.23 **	−0.27 ***	−0.21 **	−0.06	−0.24 ***	−0.07
R^2^	0.003	0.001	0.019	0.024 *	0.055 **	0.073 ***	0.044 **	0.004	0.056 ***	0.004
Age	−0.06	0.02	−0.14	−0.20 **	−0.26 ***	−0.30 ***	−0.21 **	−0.07	−0.24 **	−0.05
R/BP	0.02	0.03	−0.01	0.27 ***	0.15 *	0.21 **	0.02	0.01	−0.01	−0.14
R^2^	0.003	0.001	0.019	0.096	0.078	0.117	0.044	0.004	0.057	0.023
Δ R^2^	0.001	0.001	0.000	0.072 ***	0.023 *	0.044 **	0.000	0.000	0.000	0.019
Age	0.01	0.04	−0.07	−0.12	−0.19 **	−0.15 *	−0.12	−0.03	−0.13	−0.01
R/BP	0.02	0.04	−0.02	0.24 ***	0.14 *	0.13 *	−0.02	0.01	−0.06	−0.14
Disturbed Identity	0.15	−0.03	0.16	0.25 ***	0.17 *	0.56 ***	0.37 ***	0.02	0.33 ***	0.11
Lack of Identity	0.40 ***	0.41 ***	0.37 ***	0.05	0.34 ***	−0.11	0.06	0.29 **	−0.02	0.24 **
Consolidated Identity	−0.05	0.01	−0.02	−0.03	−0.00	0.01	0.22 *	−0.17	−0.14	0.04
R^2^	0.267	0.151	0.245	0.174	0.272	0.362	0.165	0.181	0.199	0.100
Δ R^2^	0.263 ***	0.150 ***	0.226 ***	0.078 **	0.195 ***	0.245 ***	0.121 ***	0.177 ***	0.143 ***	0.077 ***

* *p* < 0.05; ** *p* < 0.01; *** *p* < 0.001; ADP-IV = Assessment of DSM-IV Personality Disorders; SCIM: Self-Concept and Identity Measure; ED = eating disorder; R/BP = restrictive/binge-eating/purging ED subtype. PAR = paranoid; SZ = schizoid; ST = schizotypal; AS = antisocial; BPD = borderline; NAR = narcissistic; AV = avoidant; DEP = dependent; OC = obsessive–compulsive dimensional personality disorder scores.

## Data Availability

Data are available upon request at laurence.claes@kuleuven.be.

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
