# Peer review of "The Association Between Identity Functioning and Personality Pathology in Female Patients with Eating Disorders"

_nutrients, 2025, doi:10.3390/nu17142329_

Round 1

Reviewer 1 Report

Comments and Suggestions for Authors

This is a strong, ambitious paper that bridges clinically and theoretically important domains. Its key strength lies in applying a nuanced identity framework to ED-PD comorbidity. However, the manuscript would benefit from streamlining background material, more critical interpretation of findings, and a broader view on clinical applications.

Title and abstract

The title could benefit from simplification. For example, the phrase "Searching for the Self" is evocative but vague and not well explained in the abstract.

The abstract is too dense. It compresses multiple findings and hypotheses without enough flow. It would be clearer if the results were grouped per objective, and the clinical message could be reinforced in the conclusion.

Please, clarify the framing of “Searching for the Self” or revise to align more clearly with the operationalized identity constructs. Break down abstract findings in a clearer structure (aims, methods, results, conclusions). Include a brief mention of the clinical relevance of identity-based interventions already in the abstract.

Introduction

The introduction is rich in content and well-grounded in the literature. However, it is somewhat repetitive and overly extensive in reviewing past work, especially in sections 1.2 and 1.3. This may reduce the focus on the specific rationale of the current study. Reduce redundancy in the background literature on identity and EDs. Several ideas (e.g., identity confusion as a mechanism in both restrictive and binge-purge types) are repeated. The final paragraph of the introduction is quite clear and structured but could benefit from explicitly highlighting the novelty of the study (e.g., first to assess the three identity dimensions in EDs using dimensional PD outcomes).

Methods

This section is clear and appropriately structured, but there are several issues:

The sample is limited to female inpatients, but this is only acknowledged later. The implications of this limitation could be noted earlier.

The inclusion of EDNOS (47% of the sample) should be better justified. EDNOS is a heterogeneous category, and collapsing it into restrictive vs. binge/purge groups may mask clinical differences.

Results

The presentation is comprehensive and methodologically rigorous. However, it is somewhat overwhelming, with a heavy reliance on tables and few interpretative signposts.

Emphasize the most clinically relevant findings within each subsection rather than presenting all correlations with equal weight. Clarify the effect sizes and their clinical significance: not all statistically significant associations are meaningful. The predictive strength of the identity scales should be better contextualized: for example, is the additional variance explained clinically meaningful?

Discussion

The discussion effectively ties back to the hypotheses and broader theoretical frameworks. It draws important implications about identity as a transdiagnostic mechanism. Nevertheless, some sections remain overly descriptive and speculative.

Avoid reiterating the results; focus more on interpretation and implications.

The concept of “lack of relationships” vs. “dysfunctional relationships” used to distinguish PDs predicted by different identity deficits is interesting but needs clearer conceptual framing and support.

The clinical implications of the findings for treatment models other than DBT-E should be briefly discussed, especially given the study’s broader dimensional scope.

Broaden the discussion of clinical utility beyond DBT-E. Clarify whether the study suggests that identity-focused interventions should be used transdiagnostically or tailored to specific PD/ED constellations.

Limitations and Future Directions

Emphasize the limitation regarding collapsing EDNOS and the possible diagnostic heterogeneity it introduces.

Consider including a note on cultural generalizability, given the use of Dutch translations and a Flemish sample.

Author Response

Reviewer 1:

This is a strong, ambitious paper that bridges clinically and theoretically important domains. Its key strength lies in applying a nuanced identity framework to ED-PD comorbidity. However, the manuscript would benefit from streamlining background material, more critical interpretation of findings, and a broader view on clinical applications.

Title and abstract

Comment 1: The title could benefit from simplification. For example, the phrase "Searching for the Self" is evocative but vague and not well explained in the abstract.

Response 1: We have removed the phrase “Searching for the Self” from the title.

The abstract is too dense. It compresses multiple findings and hypotheses without enough flow. It would be clearer if the results were grouped per objective, and the clinical message could be reinforced in the conclusion.

Comment 2: (2a) Please, clarify the framing of “Searching for the Self” or revise to align more clearly with the operationalized identity constructs. (2b) Break down abstract findings in a clearer structure (aims, methods, results, conclusions). (2c) Include a brief mention of the clinical relevance of identity-based interventions already in the abstract.

Response 2:

(2a) We have removed the phrase “Searching for the Self” from the title.

(2b) We have broken down the abstract findings in a clearer structure as requested by the reviewer.

(2c) In the last two sentences we briefly mentioned the clinical relevance of identity-based interventions.

“The co-occurrence of identity issues in both PDs and EDs underscores the role of identity as a transdiagnostic feature. Accordingly, using identity-based interventions in treatment may have broad therapeutic benefits across these disorders.”

Introduction

Comment 3: The introduction is rich in content and well-grounded in the literature.

Thank you.

(3a) However, it is somewhat repetitive and overly extensive in reviewing past work, especially in sections 1.2 and 1.3. This may reduce the focus on the specific rationale of the current study. Reduce redundancy in the background literature on identity and EDs. Several ideas (e.g., identity confusion as a mechanism in both restrictive and binge-purge types) are repeated.

(3b) The final paragraph of the introduction is quite clear and structured but could benefit from explicitly highlighting the novelty of the study (e.g., first to assess the three identity dimensions in EDs using dimensional PD outcomes).

Response 3:

As requested by the reviewer, (3a) we have shortened the Introduction, with particular attention to Sections 1.2 and 1.3.; and (3b) have emphasized the novelty of the study.

“The present study aimed to examine the prevalence of categorical PDs in patients with ED and was the first to investigate the associations between identity functioning, as conceptualized by Kaufman et al. [4], and dimensional PD symptom scores in individuals with ED. Specifically, the study pursued five main objectives.”

Methods

This section is clear and appropriately structured, but there are several issues:

Comment 4: The sample is limited to female inpatients, but this is only acknowledged later. The implications of this limitation could be noted earlier.

Response 4: We have added the following information at the end of the participant paragraph:

Importantly, since the sample consisted exclusively of female patients with EDs, no conclusions can be drawn regarding male patients..

This limitation is also repeated in the limitation section of the discussion.

Comment 5: The inclusion of EDNOS (47% of the sample) should be better justified. EDNOS is a heterogeneous category and collapsing it into restrictive vs. binge/purge groups may mask clinical differences.

Response 5: This limitation has been acknowledged and discussed in the limitations section.

“Additionally, EDNOS represents a heterogeneous category, and collapsing it into restrictive versus binge/purge groups may mask important clinical differences.”  

Results

The presentation is comprehensive and methodologically rigorous. However, it is somewhat overwhelming, with a heavy reliance on tables and few interpretative signposts.

Comment 6: Emphasize the most clinically relevant findings within each subsection rather than presenting all correlations with equal weight. Clarify the effect sizes and their clinical significance: not all statistically significant associations are meaningful.

Response 6: In response to Reviewer 1's comment, we have added the following to the Methods section: “Hemphill [44] suggests that correlation coefficients below .20, between .20 and .30, and above .30 represent small, medium, and large effects, respectively.” Accordingly, we have emphasized the strongest correlations in our interpretation of the results, and we have applied these guidelines consistently in the Results section.

Comment 7: The predictive strength of the identity scales should be better contextualized: for example, is the additional variance explained clinically meaningful?

Response 7: We have explained the predictive strength of the identity scales in a more contextualized manner taking into account the additional variance explained by these scales.

We have added the following information in the Results section of the manuscript:

“The results of the hierarchical regression analyses with the dimensional PD scales as dependent variables are displayed in Table 3. Overall, the findings indicate that adding the three identity scales explained a significant amount of additional variance in all dimensional PD scores, above and beyond the effects of age and ED subtype. The largest increases in explained variance were observed for Cluster A Paranoid PD (ΔR² = .263), Schizotypal PD (ΔR² = .226), and Histrionic PD (ΔR² = .245). When examining the total variance explained across all PDs by age, ED subtype, and identity dimensions, the proportion of explained variance ranged from R² = .100 (Obsessive-Compulsive PD) to R² = .360 (Histrionic PD). These results suggest that there remains substantial room for improving the prediction of dimensional PDs beyond age, ED subtype, and identity dimensions.”

Discussion

The discussion effectively ties back to the hypotheses and broader theoretical frameworks. It draws important implications about identity as a transdiagnostic mechanism. Nevertheless, some sections remain overly descriptive and speculative.

Comment 8: Avoid reiterating the results; focus more on interpretation and implications. Response 8: We have shortened the repetition of the results and removed speculative interpretations (see in revised manuscript  with track changes).

Comment 9: The concept of “lack of relationships” vs. “dysfunctional relationships” used to distinguish PDs predicted by different identity deficits is interesting but needs clearer conceptual framing and support.

Response 9: Given that this interpretation cannot be fully grounded in theoretical frameworks, we have removed this sentence from the Discussion.

Comment 10: The clinical implications of the findings for treatment models other than DBT-E should be briefly discussed, especially given the study’s broader dimensional scope. Broaden the discussion of clinical utility beyond DBT-E. Clarify whether the study suggests that identity-focused interventions should be used transdiagnostically or tailored to specific PD/ED constellations.

Response 10: We have addressed the reviewer’s comments and rewritten the paragraph as follows:

“The co-occurrence of identity disturbance in both PDs and EDs underscores its role as a transdiagnostic feature. Given its central role, identity disturbance can be considered a valuable treatment target in both PDs and EDs. Schema therapy (ST), for example, addresses maladaptive schemas underlying identity disturbance, helping patients develop a more integrated and stable sense of self; it has shown efficacy for both PDs and chronic EDs [57,58]. Mentalization-based treatment (MBT), originally developed for BPD, focuses on improving reflective functioning and self-understanding, and has been adapted successfully for EDs to enhance self-structure and emotional regulation [59]. Dialectical behavior therapy (DBT) targets identity disturbance indirectly through skills modules focused on emotion regulation, mindfulness, and interpersonal effectiveness, and has demonstrated effectiveness in patients with comorbid EDs and BPD [60, 61]. By explicitly addressing identity disturbance, these approaches provide promising avenues for targeting the shared vulnerabilities underlying both PDs and EDs.”

Limitations and Future Directions

Comment 11: Emphasize the limitation regarding collapsing EDNOS and the possible diagnostic heterogeneity it introduces.

Response 11: This limitation has been acknowledged and discussed in the limitations section.

“Additionally, EDNOS represents a heterogeneous category, and collapsing it into restrictive versus binge/purge groups may mask important clinical differences.”  

Comment 12: Consider including a note on cultural generalizability, given the use of Dutch translations and a Flemish sample.

Response 12: This limitation has been acknowledged and discussed in the limitations section.

“Finally, as most participants were White and from Western backgrounds (i.e., Flemish), the findings may not be generalizable to individuals from other ethnic, cultural, or regional groups.”

Reviewer 2 Report

Comments and Suggestions for Authors

Thank you for the opportunity to review the manuscript entitled “Searching for the Self”: The association between identity functioning and personality pathology in female patients with 
eating disorders. 

The research idea is interesting and well-developed. I believe the article has high potential for publication, so please review my comments to improve the present draft. 

Abstract

  • Please remove the M and SD from the abstract.
  • ”In line with previous research” is not appropriate to be inserted into the Abstract; please develop it later in the manuscript and explain what is about.

Introduction

  • The introduction must explain the keywords and develop the knowledge. Please move paragraphs from Results into this part to better explain. There are only some ideas about the topic.
  • Please reorganize the phases os steps or hypothesis, to have a clear presentation of the methodology. This part must be eliminated from the Introduction and inserted into the Materials and Methods.

Material and methods

  • Please explain the alpha-chronbach lower scores
  • Insert all subsections conforming to the journal`s recommendations.

Results

  • The description of the respondents - the descriptive statistics is missing. Only age and sex are taken into account for the Results.
  • Are the respondents all mentally healthy with no other comorbidities (psychological, physical etc) declared or suspected?

Discussion section must be improved.

  •  

Author Response

Reviewer 2:

Thank you for the opportunity to review the manuscript entitled “Searching for the Self”: The association between identity functioning and personality pathology in female patients with eating disorders. 

Comment 1: The research idea is interesting and well-developed. I believe the article has high potential for publication, so please review my comments to improve the present draft. 

Response 1: Thank you.

Abstract

Comment 2: Please remove the M and SD from the abstract.

Response 2: We have deleted the M and SD from the abstract.

Comment 3: ”In line with previous research” is not appropriate to be inserted into the Abstract; please develop it later in the manuscript and explain what is about.

Response 3: As requested by the reviewer, we have deleted “In line with previous research” from the Abstract.

Introduction

Comment 4: The introduction must explain the keywords and develop the knowledge. Please move paragraphs from Results into this part to better explain. There are only some ideas about the topic.

Response 4: We are certainly willing to follow the suggestions of Reviewer 2; however, it is not entirely clear to us which elements of the Results section should be included in the Introduction. Could the reviewer please provide more specific information about his/her expectations?

Comment 5: Please reorganize the phases or steps or hypothesis, to have a clear presentation of the methodology. This part must be eliminated from the Introduction and inserted into the Materials and Methods.

Response 5: Until now, we have always presented our research questions and hypotheses at the end of the Introduction section when publishing in Nutrients. Could the reviewer please clarify whether they would like us to move the research questions to the Materials and Methods section instead? Thank you.

Material and methods

Comment 6: Please explain the alpha-chronbach lower scores

We added the following paragraphs in the Materials section:

Response 6a (SCIM):

“According to George and Mallery [42], alpha values above .70, .80, and .90 indicate acceptable, good, and excellent reliability, respectively. The Disturbed Identity and Lack of Identity scales demonstrated good reliability coefficients, while the Consolidated Identity Scale exhibited marginally acceptable reliability. These results align with previous studies employing the Dutch version of the SCIM [34,39]. Excluding the item “I am basically the same person that I’ve always been” from the Consolidated Identity Scale would have increased the alpha coefficient to α = .72, indicating acceptable reliability. However, to ensure comparability with prior findings [34,39], this item was retained.”

Response 6b (APD-IV):

Internal consistency (Cronbach’s alpha) for the ADP-IV dimensional scales ranged from .68 (marginally acceptable reliability; Antisocial PD) to .81 (good reliability; Schizotypal PD). Excluding the item “I experience little or no feelings of guilt or remorse when I have done something wrong” from the Antisocial PD scale would have increased the alpha coefficient to α = .71, indicating acceptable reliability. Nevertheless, this item was preserved to allow for direct comparison with earlier studies [34,39].”

The lower reliability of both scales has also been acknowledged as a limitation of the present study.

Comment 7: Insert all subsections conforming to the journal`s recommendations.

Response 7: We believe that all required subsections have been included in the Materials and Methods section. Please let us know if there are any specific subsections you would like us to add. Thank you.

Results

Comment 8: The description of the respondents - the descriptive statistics is missing. Only age and sex are taken into account for the Results.

Response 8: Based on the available data, we were only able to include information on the educational level of the participants.

“Regarding the highest level of education, 5 (2.8%) patients had completed primary education, 50 (28.4%) secondary education, and 121 (68.8%) higher education.”

We have acknowledged this as a limitation of the present study and have added the following sentence to the limitations section of the Discussion.

“Furthermore, we did not have access to patients' medical files and therefore lacked information on detailed sociodemographic variables as well as physical and psychological comorbidities, which should be addressed in future studies.”

Comment 9: Are the respondents all mentally healthy with no other comorbidities (psychological, physical etc.) declared or suspected?

Response 9: We did not have permission to access the patients' medical files. ED symptoms were assessed using the EDI-3 Symptom Checklist, and PD symptoms were assessed using the ADP-IV questionnaire. However, we agree with the reviewer that information on physical and psychological comorbidities is important, and we have acknowledged the lack of this information as a limitation of our study.

We have added the following sentence to the limitations section of the manuscript.

“Furthermore, we did not have access to patients' medical files and therefore lacked information on detailed sociodemographic variables as well as physical and psychological comorbidities, which should be addressed in future studies.”

Comment 10: Discussion section must be improved.

Response 10: We have made further revisions to the Discussion section based on the feedback provided by Reviewer 1. However, we would like to invite Reviewer 2 to specify which aspects of the Discussion still require improvement. Thank you.

Reviewer 3 Report

Comments and Suggestions for Authors

The manuscript “Searching for the Self”: The association between identity functioning and personality pathology in female patients with eating disorders is interesting and of value. The aim of research was to investigate the associations between the three identity dimensions (Consolidated Identity, Disturbed Identity, Lack of Identity) and symptoms of personality disorders (PDs) in 176 female inpatients with an eating disorder.

The manuscript is well crafted. However, it needs some minor revisions.

Abstract: well done

Introduction: very substantive and thorough review

Material & Methods: well done

Results: Table 2, it is unclear how age and ED subtype were controlled in these calculations, please explain what ED subtype was included here.

Discussion: It is very thorough discussion. A very valuable transdiagnostic approach was highlighted.

My concern: part 4.3 Associations between Dimensional SCIM Identity Scale Scores & Age and ED Subtype, please comment the associations with ED subtype, only age was discussed here.

Author Response

Reviewer 3:

The manuscript “Searching for the Self”: The association between identity functioning and personality pathology in female patients with eating disorders is interesting and of value. The aim of research was to investigate the associations between the three identity dimensions (Consolidated Identity, Disturbed Identity, Lack of Identity) and symptoms of personality disorders (PDs) in 176 female inpatients with an eating disorder.

Thank you!

The manuscript is well crafted. However, it needs some minor revisions.

Comment 1: Abstract: well done

Response 1: Thank you!

Comment 2: Introduction: very substantive and thorough review

Response 2: Thank you!

Comment 3: Material & Methods: well done

Response 3: Thank you!

Comment 4: Results: Table 2, it is unclear how age and ED subtype were controlled in these calculations, please explain what ED subtype was included here.

Response 4:  We have calculated partial correlations between the identity dimensions and the PD dimensions controlling for age (continuous variable) and ED subtype (ED-R vs. ED-BP: dummy coded variable).  We have changed the title of Table 2 and explained the abbreviations (ED-R vs. ED-BP) below Table 2.

Table 2. Partial correlations between APD-IV and SCIM dimensional scores controlled for age and ED subtype (ED-R vs. ED-BP).

Comment 5: Discussion: It is very thorough discussion. A very valuable transdiagnostic approach was highlighted.

Response 5: Thank you!

Comment 6: My concern: part 4.3 Associations between Dimensional SCIM Identity Scale Scores & Age and ED Subtype, please comment the associations with ED subtype, only age was discussed here.

Response 6: As requested by Reviewer 3, we have added the following information in part 4.3 in the discussion.

“Unlike Rohde et al. [18], we did not observe significant differences in SCIM dimensions between ED subtypes (ED-R vs. ED-BP). This divergence may be due to differences in subgroup definitions: our ED-BP group included both BN and AN-BP patients, possibly diluting subtype-specific effects.”

Round 2

Reviewer 1 Report

Comments and Suggestions for Authors

Thank you for your work 

Reviewer 2 Report

Comments and Suggestions for Authors

The authors took into consideration all comments and other recommendations. I think that the present manuscript has improved considerably.